# Metabolic plasticity drives mismatches in physiological traits between prey and predator
Flavio Affinito [1,2,3] ✉, Rebecca L. Kordas[1], Miguel G. Matias [4,5] & Samraat Pawar [1]

Metabolic rate, the rate of energy use, underpins key ecological traits of organisms, from development and locomotion to interaction rates between individuals. In a warming world, the temperature-dependence of metabolic rate is anticipated to shift predator-prey dynamics. Yet, there is little real-world evidence on the effects of warming on trophic interactions. We measured the respiration rates of aquatic larvae of three insect species from populations experiencing a natural temperature gradient in a large-scale mesocosm experiment. Using a mechanistic model we predicted the effects of warming on these taxa's predator-prey interaction rates. We found that species-specific differences in metabolic plasticity lead to mismatches in the temperature-dependence of their relative velocities, resulting in altered predator-prey interaction rates. This study underscores the role of metabolic plasticity at the species level in modifying trophic interactions and proposes a mechanistic modelling approach that allows an efficient, high-throughput estimation of climate change threats across species pairs.

Climate change is modifying environments on a global scale, forcing species to move, adapt or go extinct[1,2]. Warming in particular is a major threat to ecosystems[2] as increased temperatures are expected to shape ecological processes, from individual metabolism[3,4] to ecosystem dynamics[5,6]. Rapid increases in global temperatures are already affecting a wide range of biological traits and ecological processes that depend on temperature[4,7], having a direct impact on species and the interactions between them[8,9]. Yet, the effects of increasing temperatures on biological traits and species interactions are often hard to predict as species respond in different ways[10,11]. Nevertheless, a mechanistic understanding of the role of temperature in driving biological rates and interactions remains to be developed through metabolic trait-based approaches[12–14].

Whole-organism metabolic rate follows a unimodal relationship with temperature (the thermal performance curve, TPC)[15–18]. A sufficiently large change in environmental temperature (and therefore body temperature in ectotherms) induces a change in metabolic rate, detectable through an elevated respiration rate, unless the temperature increases beyond an organism's thermal optimum. Respiration rates increase in response to an increase in the demand for ATP[19], requiring that individuals consume more oxygen per unit time[20,21]. This in turn changes resource demand as well as the ability of individuals to find resources, ultimately affecting growth and predation rates[20–22]. A change in resource acquisition rates then affects survivability of individuals and reproduction chances, therefore affecting individual fitness outcomes[12]. The TPC of respiration rate itself is expected to change within and across generations in response to new thermal conditions[23] (Fig. 1a, b) which would then in turn change species interactions and population dynamics[5,13,24–26]. This change in the TPC is typically seen in the form of either a vertical shift in performance (change in the normalisation constant), change in thermal sensitivity (activation energy), or both[27]. The TPC parameters ($B_0$, $E$, $E_d$; Fig. 1) can thus be compared to identify how individuals are responding physiologically to changing temperatures. For example, thermal sensitivity may change as the TPC becomes 'steeper' due to an increase in respiration rate at higher temperatures without a loss of performance at cooler temperatures over the operational temperature range (OTR, the range of temperatures experienced by a species in its natural environment, Fig. 1b[27,28]). Recent empirical evidence suggests that such changes are dependent on species-specific abilities to adapt or acclimate to new thermal conditions[27,29,30].

If the shape of the TPCs of interacting species change relative to each other, this can result in mismatches in performance (Fig. 1c, d), for example between predators and their prey[13,31]. As respiration rates change in response to temperature shifts, these mismatches in performance will modify other

[1]Imperial College London Silwood Park, Buckhurst Road, Berks SL5 7PY, UK. [2]McGill University Department of Biology, 1205 Dr Penfield Ave, Montreal, QC H3A 1B1, Canada. [3]Québec Centre for Biodiversity Science, 1205 Dr Penfield Ave, Montreal, QC H3A 1B1, Canada. [4]Museo Nacional de Ciencias Naturales (CSIC), C. de José Gutiérrez Abascal, 2, Chamartín, 28006 Madrid, Spain. [5]Rui Nabeiro Biodiversity Chair, MED—Mediterranean Institute for Agriculture, Environment and Development, University of Évora, Pólo da Mitra Apartado 94, 7006-554 Évora, Portugal. ✉e-mail: flavio.affinito@mail.mcgill.ca

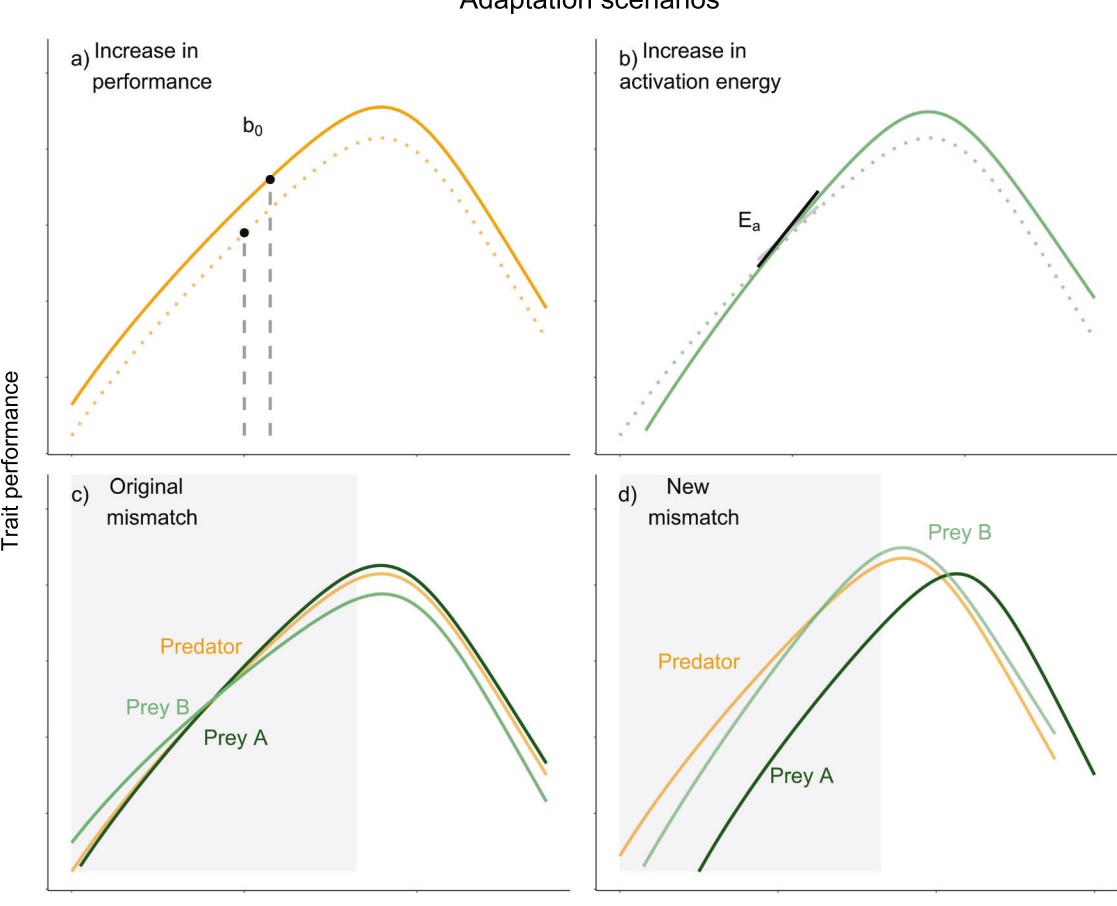

**Fig. 1 | Plasticity of TPCs may lead to mismatches in performance between interacting species.** Metabolic rates can be modelled as an unimodal function of temperature determined by biochemical processes (enzyme activation ($E$) and deactivation ($E_d$) energies) and biological parameters ($T_{pk}$ and $B_0$) from Eq. 1[92]. We estimated $B_0$ at the mean habitat temperature, representing the average baseline performance at a given site. The TPC for a species from a cooler habitat (dotted lines) may change in different ways when it has adapted to a warmer habitat (solid lines): for example, (**a**) a vertical shift in performance would increase $B_0$ or (**b**) the sensitivity of the curve ($E$) could change. In a habitat with multiple interacting species, predator (orange) and prey (greens) may have similar TPCs in cooler habitats (**c**). However, if predator and prey adapt to higher temperatures in different ways (e.g., predator adapts via an increase in $B_0$ (scenario (**a**)) and prey adapts via decrease in $B_0$ scenario (**a**) or via scenario (**b**)), then their TPCs may become further mismatched (**d**). The mismatch caused by metabolic plasticity is particularly important within the operational temperature range of species (grey area). Adapted from[7,12,13,23].

metabolic traits such as locomotion and growth[32,33]. In turn, these trait mismatches will modify predator-prey dynamics[24,25]. Such individual level changes have significant ecological implications, increases in predator search rates can lead to higher catch rates and reductions in prey species numbers leading potentially to extinction and/or prey-switching[34]. These population level changes in turn alter community dynamics, potentially rewiring food webs, altering the strengths of relationships between species and lowering the overall trophic level[9]. At the ecosystem level, these effects can modify ecosystem functioning reduce resilience and make ecosystems more prone to collapse[35]. Recent theoretical developments provide a framework to investigate how complex processes emerge from species' individual metabolic responses to new thermal conditions using mechanistic models linking metabolism to locomotion to search rates. While some empirical data have emerged over the last decade to test these theoretical predictions[36–38], both lab and field data remain scarce[39].

In this study, we investigated how the TPCs of respiration rate of three common European invertebrate species change across a thermal gradient in a large scale mesocosm experiment located in the Iberian Peninsula, and what effects these changes are likely to have on the predator-prey interactions among them. We integrate empirical measurements of respiration rates with biomechanical and consumer-resource theories to connect metabolic rates with predator search and interaction rates[13,40]. Our approach

attempts to link basal metabolic measurements with higher level temperature-dependent traits using mechanistic models of thermal performance and energetics to predict the effect of thermal acclimation or adaptation (phenotypic changes resulting from an organism coping with a new thermal regime or due to selection of better thermally adapted phenotypes in a population[12])—henceforth referred to as 'metabolic plasticity', following ref. 27. Specifically, we ask how metabolic plasticity changes predator-prey interactions, and what role differential changes in the TPCs of interacting species play in such changes. Our work emphasises the value of a mechanistic understanding of the effects of temperature on traits such as respiration, locomotion and predator search rate.

## Results
We observed contrasting patterns of change in respiration rate TPCs of *Chironomus* spp. compared to *S. striolatum* and *C. dipterum*. For *Chironomus* spp. the most elevated curve (highest $B_0$) was that of the coolest site, Toledo, followed by the intermediate site, Porto and the warmer site, Evora (Fig. 2a), consistent with the down-regulation of metabolic rates reported across organisms[41,42]. For *S. striolatum* and *C. dipterum*, the most elevated curve was that of the warmer site, Evora and the lowest curve that of the coolest site, Toledo (Fig. 2b, c). However, the model for *S. striolatum* in Toledo had large uncertainty bounds after 25 °C that overlapped with the

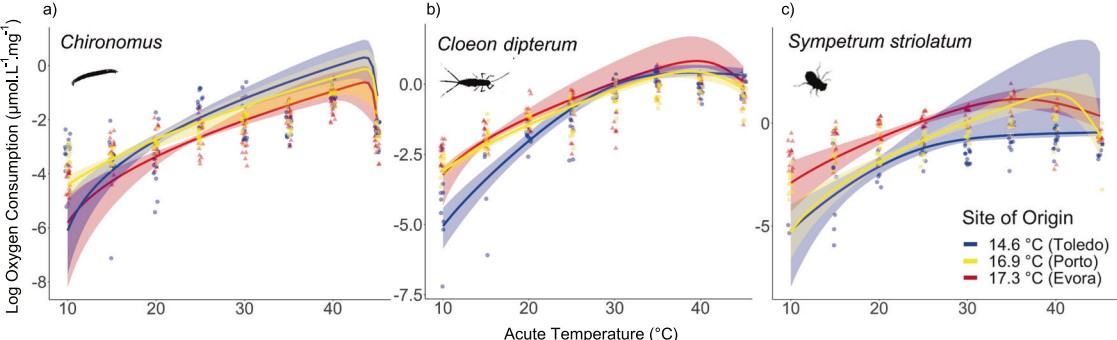

**Fig. 2 | Temperature-dependence of respiration rate varies across sites.** Mass corrected metabolism temperature performance curves with 95% confidence interval for (**a**) *Chironomus* spp., (**b**) *C. dipterum* and (**c**) *S. striolatum* were modelled using the Sharpe-Schoolfield model (Eq. 1). Data were collected from between 75 and 164 individuals for each curve (Supplementary Table 2). Note that the patterns in metabolic plasticity are similar to those proposed in Fig. 1a, b are observed here.

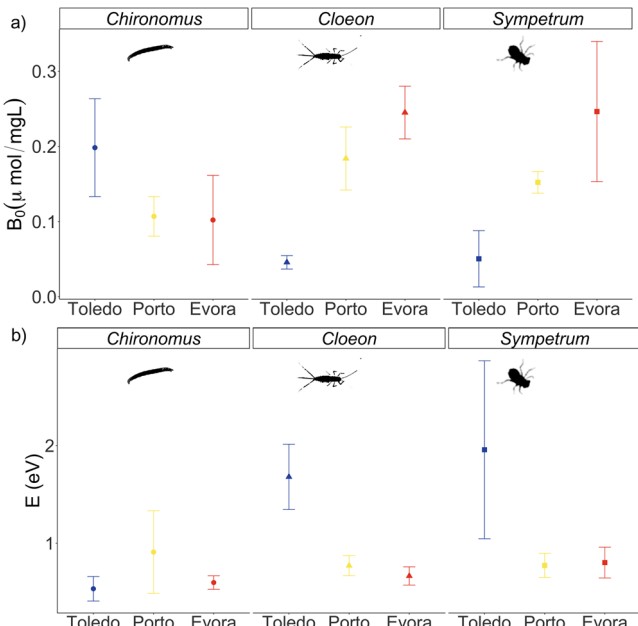

**Fig. 3 | Parameter estimates of respiration TPCs for each species from each site.** **a** Baseline performances and (**b**) activation energies were estimated from the Sharpe-Schoolfield model fitted to respiration rates. Error bars are 95% confidence intervals. Note that we observed greater variation within species for baseline performance than activation energy.

other two sites (Fig. 2c) due to a poor estimation of the activation energy parameter, $E$ (Fig. 3b). Model fit in all cases was $0.56 < R^2 < 0.82$ with almost all parameters achieving significance levels ($p < 0.05$, Supplementary Table 2). These differences in TPCs revealed a species-specific response in elevation or sensitivity, as predicted by theory (Fig. 1a, b). Specifically, the elevation parameter, $B_0$, decreased from cool to warm for *Chironomus* spp. but increased for *S. striolatum* and *C. dipterum* (Fig. 3a). However, the sensitivity parameter, $E$, did not change for *Chironomus* spp. but decreased from the cooler site to the two warmer sites for *S. striolatum* and *C. dipterum* (Fig. 3b).

Comparing predator-prey velocity TPCs revealed a predicted mismatch in performance (Fig. 1) for the *S. striolatum-Chironomus* spp. pair but not for the *S. striolatum-C. dipterum* pair (Fig. 4). In the case of *Chironomus* spp. as prey, the reduction in elevation of the prey's velocity TPC led to a greater difference in relative velocities from the coolest to warmest sites with no overlap of the curves over the operational temperature range (Fig. 4a, c), meaning that the predator is predicted to move faster than its prey in warmer sites. In contrast, in the case of *C. dipterum* as prey, velocity

TPC curves overlapped over the OTR in all three sites (Fig. 4d–f), except at temperatures below 20 °C at the intermediate site where the prey TPC was higher than its predator's (Fig. 4e). Thus, relative velocities of interacting pairs of *S. striolatum* and *C. dipterum* were predicted to remain unchanged with warming.

Modelled search rates increased with temperature at all three sites and for both foraging strategies (Fig. 5a) although the strength of that increase varied ($0.74 < E < 1.96$; Supplementary Table 3). Predator search rates were most thermally sensitive in the coolest site (Fig. 5a) although these results were driven by the large and more uncertain values of $E$ estimated in the respiration TPC fits (Fig. 3b). Thermal sensitivity was higher at the warmest than intermediate site (Fig. 5b, c). In all three sites, search rates and their sensitivity, $E$, were higher for *Chironomus* spp. (the sessile 2D strategy, driven only by predator TPC parameters) than for *C. dipterum* (the active 3D strategy, driven by both predator and prey TPC parameters, Fig. 5). Differences in search rates between prey species were smaller over the OTR than at higher temperatures. Predator search rates for *C. dipterum* prey were lower at all sites and for all temperatures than search rates for *Chironomus* spp.

## Discussion

We identified patterns of change in the thermal physiology of three ectotherm species across an environmental gradient and modelled how these produce mismatches in the temperature dependence of body velocity between species and ultimately, interaction rates between them. We found support for one hypothetical scenario of adaptive or acclamatory change—a vertical shift of the whole TPC with warming—but the direction of such shifts was species-specific. The second scenario, a change of sensitivity in the TPC, was less well supported but sensitivity tended to decline from the coolest to the warmer sites for two out of three species. By calculating velocity estimates from respiration rates, we found that there is a mismatch in trait performance in the *Chironomus* spp.-*S. striolatum* pair but not in the *C. dipterum-S. striolatum* pair. This mismatch increased in warmer sites and translated to higher predator search rates for *Chironomus* spp. than for *C. dipterum*. Finally, we found that changes in the metabolism in these species result in a decrease in thermal sensitivity of predator search rates in warmer sites.

Warming had a negative effect on the baseline performance of *Chironomus* spp. A decrease in the elevation of the TPC of *Chironomus* spp. respiration rates with warming translates to less availability of energy for all biological processes for individuals living in warmer environments. Laboratory studies of European *C. riparius*, a common *Chironomus* spp. in Western Europe, have revealed that mutation rates in the species are correlated to mean annual temperature[43] and detected thermal adaptation at the genetic level[44]. Strong selection pressure from temperature increases caused an adaptive response in as little as three generations[45], corresponding exactly to the number of generations in our experimental mesocosms.

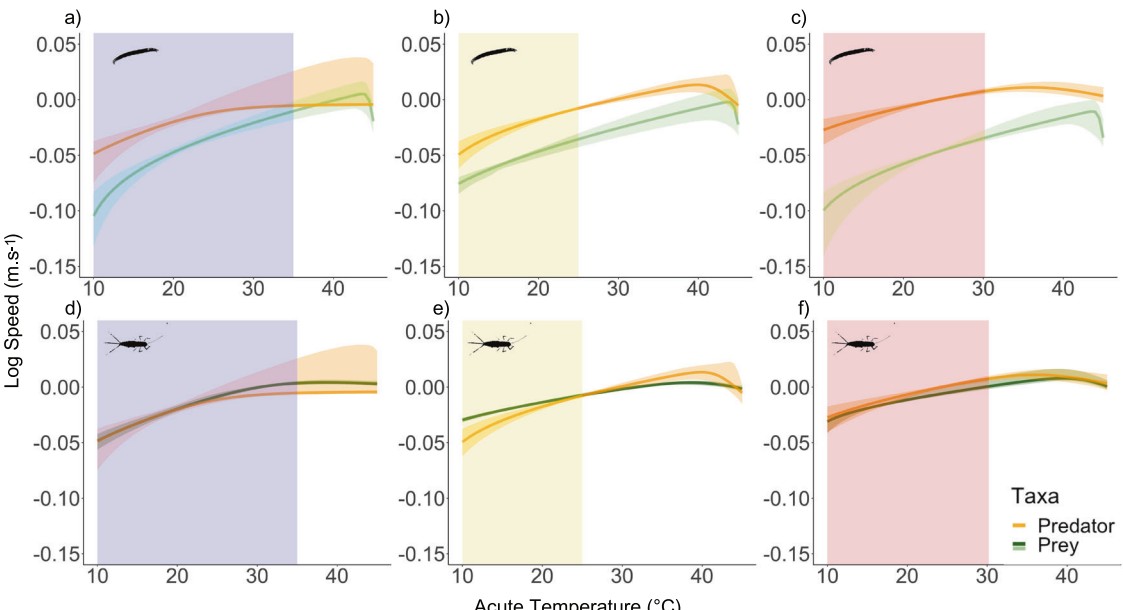

**Fig. 4 | Mismatches between species in velocity TPCs are affected by species metabolic plasticity.** Predator-prey velocity curves for Toledo (14.9 °C; (**a**) and (**d**)), Porto (16.3 °C; (**b**) and (**e**)) and Evora (17.3 °C; (**c**) and (**f**)) for each predator-prey pair (**a–c**) for *S. striolatum-Chironomus* spp. and (**d–f**) for *S. striolatum-C. dipterum*) with 95% confidence interval estimated from respiration rates (Eq. (3)). Coloured rectangular areas correspond to the range of recorded temperatures (operational temperature range) of the sites (Supplementary Fig. 1). Note that species specific metabolic plasticity, such as decrease in $B_0$ (*Chironomus*), result in changes in mismatch magnitude over the OTR.

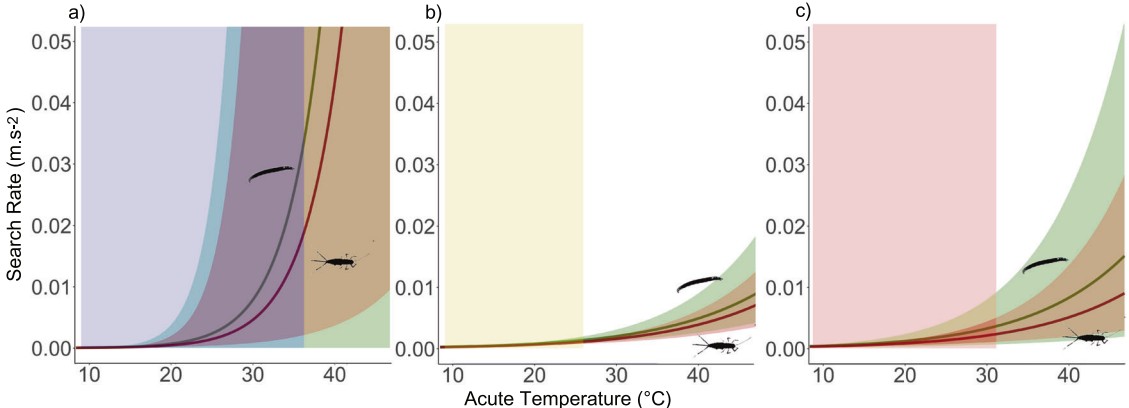

**Fig. 5 | Predator search rates are less thermally sensitive in warmer habitats.** Predicted predator search rate models for *Chironomus* spp (green) and *C. dipterum* (red) in (**a**) Toledo, (**b**) Porto and (**c**) Evora. Curves are derived from Eq. (5) (sessile prey strategy) for *Chisonomus* spp and Eq. (4) (active capture strategy) for *C. dipterum*. Search rates for *Chironomus* spp are higher than for *C. dipterum*, especially at higher temperatures. Strategies correspond to the most biologically relevant model for the interacting pair, based on observations of insect movement in nature. Coloured rectangular areas correspond to the range of recorded temperatures (operational temperature range) of the sites (see Supplementary Fig. 1).

Hence, it is possible that *Chironomus* spp. in our experiments lack the thermal plasticity to acclimate to the increase in temperature[45] and are failing to effectively adapt to the new thermal conditions. Indeed, the decrease in elevation of the respiration rate TPC suggests an inability to meet the increased need for oxygen in warmer conditions for *Chironomus* spp[12], with knock-on effects on emergent traits. Thus, velocity TPCs of *Chironomus* spp. were consistently lower than those of its predator *S. striolatum* throughout their operational temperature range. In fact, this mismatch got larger in warmer sites, a result previously found for another dragonfly species foraging on tadpoles[11]. Indeed, as *S. striolatum* respiration TPCs increase in elevation with warming, individuals in warmer sites will have more energy available to allocate to other traits such as movement. This in turn had an effect on search rates as the predator becomes better able to forage for its prey[46]. With predators able to move relatively faster than its prey in warmer environments, *Chironomus* spp. may find itself unable to

escape predation and become a preferred prey item. That is, the inability of *Chironomus* spp. to acclimate rapidly enough to warming may hinder its ability to swim away and avoid predators. This may be counteracted by the fact that *Chironomus* spp. can bury themselves in the sediment to escape detection[47], reducing the need to acclimate to warming.

TPCs for *C. dipterum* and *S. striolatum* increased in elevation with warming but decreased in sensitivity, suggesting that there may be a trade-off, where increasing overall performance results in a reduction in thermal niche breadth. Few studies on *S. striolatum* physiological responses to temperature are available[48,49] but work on *C. dipterum* has revealed a highly plastic response to warming and predation[50,51] as well as a high acclimation potential to temperature[52]. Metabolic plasticity is not uncommon in invertebrates exposed to chronic warming[27], and it is possible that these two species have acclimated to warming by elevating their TPCs at the cost of reduced thermal niche breadth. Indeed, similar patterns of acclimation in

respiration TPCs for *S. striolatum* and *C. dipterum* translated to no change in the mismatch of velocity TPCs where relative velocity of prey and predator did not change with warming. Similarly, search rates for *C. dipterum* were lower than for *Chironomus* spp. because the absence of a mismatch in velocity makes it relatively more difficult for *S. striolatum* to forage for *C. dipterum*. The elevated search rates predicted in the cooler site were due to the elevated sensitivity of *S. striolatum* respiration TPCs and may be due to its high parameter uncertainty. In contrast to *Chironomus* spp., *C. dipterum* is not known to be able to hide under the sediment from *S. striolatum* in its environment. Both species are found swimming in the water column where there is no abiotic refuge for the prey. Thus, *C. dipterum* can only avoid its predator by swimming faster to get away, or vice versa for *S. striolatum* to catch its prey, which may explain similar acclimation patterns for this interacting pair.

Our predictions of mismatches in traits for both predator-prey pairs in this study are consistent with the biology of these species and predictions from more general theory, and show that dimensionality and differences in acclimation to warming interact to modify interaction rates[13,40,53]. Furthermore, they support the growing body of literature emphasising that the response of predation to temperature can vary quite significantly with species identity[10], size[54] and movement (foraging or avoidance) strategy[33]. Indeed, only by accounting for the divergent life histories and adaptive capacities of both prey taxa can we explain the different patterns in velocity mismatch and predator search rates observed. This further highlights the need to account for the temperature dependence of both prey and predator traits when studying mismatches in performance rather than the widespread focus on predators only[29]. In particular, quantifying prey-specific TPCs of metabolic rate and emergent traits to temperature, and other, non-temperature related traits such as detection rate of chemical cues[30], are essential to predict interaction rate accurately as these will further modulate mismatches in trait performance between predators and prey.

A focus on metabolism and how it scales up to higher order (emergent) traits can help fill the empirical gap in understudied species such as freshwater invertebrates. Indeed, in general, very few studies have experimentally measured the TPCs of velocity due to the inherent challenges of capturing the body speed of individuals at different temperatures under realistic conditions[55–58]. Measuring metabolic rate is more feasible, and our results here highlight how, when combined with mechanistic modelling, this can be used to predict biologically meaningful mismatches in predator-prey velocities and their impact on interactions. Future studies should also consider the effect of variation in individual behaviour that may temporarily override metabolic constraints (e.g. a short burst of high swimming speed by the predator followed by an inactive recovery period).

Time constraints prevented us from empirically testing our predictions about the consequences of warming and plasticity on predator-prey performance mismatches and the emergent temperature-dependence of interaction rates. Testing predictions of such mechanistic theory for temperature-dependent predator-prey interactions would require functional response experiments using the same species-pair and site combinations[25,46,59,60]. The search rates estimated from these experiments could then be compared with the predicted values (Fig. 5). However, detecting acclimatory changes would require a higher level of replication to estimate the thermal plasticity of search rate parameters. Furthermore, the predictions of the model for the temperature-dependence of velocity would also need to be tested (Fig. 4). Direct measurement of body velocity can be done using video tracking powered by machine learning[61–65]. This is challenging because of the difficulty of automated tracking in such environments, which likely explains the paucity of existing data on the movement of freshwater macroinvertebrates[66,67]. Lab experiments with wild-caught organisms may help address this gap[68,69] provided the challenge of recreating the natural thermal and structural environment can be overcome.

Future work should also consider some of the limitations of the assumptions of the consumer-resource theory underlying our modelling here. First, this theory assumes simple brownian motion of both predator and prey individuals. This has been shown to be a reasonable approximation for obtaining empirically-relevant predictions for a range of animal types[5]. Also, while other, more intricate random movement strategies such as Levy flights have been proposed, support for them is mixed[70–73]. Yet, the existence of levy flights in small animals, and especially in freshwater invertebrates has not been tested. Second, the effect of dimensionality is largely dependent on the environment and the predator's detection apparatus, as minimum detection distance depends both on the environmental medium (e.g. air vs. water) and modality (e.g., visual vs. olfactory)[40]. Third, here we used a very general model for locomotion in water and cost of transport[74–76] because more specific, empirically validated models based on the specific locomotion techniques and physical constraints experienced by freshwater invertebrates are not available[77,78]. This is another empirical and theoretical gap that future studies will need to address to make such mechanistic models for predicting interaction rates more accurate. Finally, for simplicity, and because no information was available on the energy budgets of our study organisms, we assumed that all the energy produced by metabolism was invested in locomotion at the short timescales of interaction. Accounting for the partitioning of metabolic energy to growth and maintenance at longer timescales[79] as well as muscular inefficiency (up to 60% in animals)[74,77] would also refine model predictions. As such, the elevation of the predicted TPCs of velocity here are likely overestimates.

In the specific context of macroinvertebrates in the Iberian Peninsula, our findings suggest that the predatory *S. striolatum* can cope with rising temperatures, potentially benefiting from altered prey selection behaviours in a warming climate. Switching to *Chironomus spp.* may have effects on the local ecosystem, however, as midge larvae are already an important prey item for many species and play a major role linking producers and consumers by feeding on the benthos and moving many inorganic nutrients, as well as carbon, into the food web[80,81]. Decline in chironomid numbers may therefore lead to overall declines in energy availability in freshwater systems with negative consequences for higher trophic level organisms.

In summary, our study shows how mechanistic modelling of metabolic rates can be used to identify and model species-specific mismatches in velocity that affect predator search rates at this larval stage in three widespread European invertebrate species. As global temperatures rise, ectothermic species, especially those in freshwater systems, are already facing increasing challenges[82]. This is especially true for aquatic species, particularly in freshwater systems[83]. Furthermore, few studies on invertebrate species' responses to changing temperatures focus on larval stages even though these may be the most susceptible to warming. Therefore, despite its theoretical limitations, our study shows that valuable insights can be obtained into the metabolic challenges faced by aquatic predators and prey. And because it is relatively feasible to obtain estimates of metabolic rate across diverse species in such environments compared to more emergent traits such as body velocity, a relatively coarse but general mechanistic modelling approach like this can allow an efficient, high-throughput estimation of climate change threats across diverse species pairs, that can also taxa and interactions of particular concern to be identified, after which more detailed empirical and theoretical work can be conducted on them. For example, our results identify some species that will be able to cope (*C. dipterum*) with warming, whilst others may not (*Chironomus* spp.). This preliminary information can guide further experiments that combine metabolic rate measurements with those of velocity to more accurately predict species- to community-level responses to warming by accounting for the effects of metabolic mismatches in not just the thermal responses of functional traits, but also in their rates of acclimation in the face of changing temperatures.

## Methods
### Study sites
We focused on three sites in the Iberian Peninsula that varied in altitude and latitude (Supplementary Fig. 1). At each site, 32 artificial mesocosm ponds (each 1000 L when full) were seeded with an assemblage of freshwater species collected from nearby water bodies (including lakes, ponds and small streams; see description in[84]) during autumn-winter 2014–2015. The

communities in the mesocosms were then left to assemble naturally over the following 2 years. Sampling individuals from mesocosms avoided any effects of habitat heterogeneity from natural ponds on the individuals sampled and enabled comparison between and within sites as all mesocosms experienced the same conditions in each site. Most invertebrate species found in the ponds are efficient dispersers upon reaching adult stage and were found throughout ponds within a site and across sites[84]. We sampled the ponds during the local spring season at each site in April 2017. Mesocosm temperatures were recorded with submerged loggers every hour starting when each site was constructed up until the end of our sampling period in April 2017. We isolated the period of temperature recordings that corresponded to the aquatic larval stage of our focal taxa (August 2016–April 2017), as those were the conditions that the individuals acclimated to prior to our experiments. The sites experienced different thermal regimes (Supplementary Figs. 2 and 3) in terms of mean temperatures (±95%CI): Toledo (14.56 ± 0.85 °C), Porto (16.95 ± 0.80 °C) and Évora (17.30 ± 0.76 °C). Due to the nature of the mesocosm set up, the main drivers of environmental differences across regions were temperature and precipitation[84]. Whilst it is possible that precipitation affected other parameters, such as nutrient concentration, these effects would have been mainly reflected in population abundances rather than in metabolic rate measurements of individuals that are mainly dependent on temperature and body mass[17,22].

## Study organisms

We measured respiration rates of the aquatic larvae of three insect taxa from the Iberian Peninsula: nymphs of the dragonfly, *Sympetrum striolatum* and two of its prey items the larvae of *Chironomus* spp. (a genus of midges) and *Cloeon striolatum* (a mayfly). Individuals were collected with 500 μm mesh nets from freshwater mesocosms and were identified to genus or species level using a dissecting microscope, following experiments and preservation in 70% ethanol. Individuals were sampled from across most mesocosms at each site and were pooled together for respiration trials. We targeted species based on trophic level (predator vs. prey) and abundance at each site (minimum 100 individuals per site). To identify trophic relationships between taxa, we carried out overnight feeding trials (see Supplementary Methods 1).

We determined the foraging strategies of each predator-prey pair based on the morphology and locomotory behaviour of each prey taxa[13,38]. In their aquatic stage, *S. striolatum* (predator) and *C. dipterum* (prey) are free-swimming pelagic species[85,86]. In contrast, *Chironomus* spp. are benthic and found mostly living in the sediment[87]. When prey velocity was likely to be negligible relative to that of the predator, we classified the interaction as a 'sessile prey' strategy (as with *S. striolatum* and *Chironomus* spp[40]). In contrast, when prey velocity was non-negligible and likely to affect the predator's ability to capture prey, we defined the foraging strategy as 'active capture' (as with *S. striolatum* and *C. dipterum*). These contrasting prey behaviours are expected to produce different interaction strengths, depending on the environmental space they occur in.

## Metabolic rate measurements

We measured respiration rates as a proxy for metabolic rate, following the methods of ref. 88. For each population, we measured individual oxygen consumption rates at acute temperatures spanning 10–45 °C at 5 °C intervals. Such a wide range of temperatures was chosen to capture the full unimodal form of the TPCs and reduce bias in the estimation of the thermal sensitivity parameter[89].

Prior to measurements, individual insects were stored for 24 h in containers filled with mesocosm water that had been filtered using 20 μm mesh nets and held at ambient temperature and in natural light conditions. This allowed for gut clearing of all individuals to reduce the confounding effect of digestion on metabolic rate. Individuals were acclimated to acute experimental temperatures for 15 min prior to respiration measurements to avoid a shock response as follows. Each individual was placed in a plastic container filled with 100 ml of filtered mesocosm water at ambient temperature, which was then placed in a water bath set to the acute temperature.

This allowed the water inside the container to reach the acute temperature in ~15 min.

For each respiration trial, eight glass vials (capacities 0.75 mL or 2 mL) were filled (leaving no head space) with filtered air-saturated water from the local mesocosms, which was kept at the experimental temperature. A magnetic stir-bar maintained mixing of the water column in each glass vial, separated from the organism by a small mesh screen placed at the bottom of the vial to minimise any additional stress to the organism. A single individual was placed into each of seven glass vials, while the eighth vial contained filtered water only and thus served as a control to account for sensor drift and the oxygen changes due to microorganisms in the mesocosm water. All vials were then sealed, placed into a magnetic stir-bar rack, and fully immersed in a water bath set to the experimental temperature. Each individual was only assayed once.

Oxygen concentration was measured during three periods of ~30 s each (logging every second) using an oxygen microelectrode (MicroResp, Unisense, Denmark) fitted through a capillary in the lid of each vial[88]. All trials were ended when oxygen concentrations were reached 70% of the starting value, to minimise stress to the insects. The oxygen consumption rate of each individual was calculated as the slope coefficient of a linear regression fitted to all of the data points. For each trial, we corrected for differences in vial volume and in the oxygen concentration of the control vial. The body length of each individual was measured using a dissecting microscope and converted to dry mass using length-weight allometries measured for taxa from our sites (Supplementary Table 1).

## Modelling

Our overall objective was to use established mechanistic models of metabolism, movement, and predation to estimate the temperature dependence of unmeasured velocities and search rates for our species from measured respiration rates. These estimates could then be used to compare mismatches in interaction (relative velocity, search rate) traits that were not measured but that are dependent on the measured metabolic rate.

Organisms consume oxygen (i.e. respiration rate) for aerobic respiration to produce ATP for energy[21], this ATP is then used for many cellular purposes from maintenance to growth, to reproduction and movement[4,12]. In fact, a significant part of the energy produced by metabolism (i.e. respiration) is allocated to muscle contraction and locomotion, whilst the rest goes to growth, maintenance, and reproduction[77]. It is well-established that there is a relationship between the energetic cost of transport (the amount of energy required for movement) and metabolism which varies by locomotion type[26,74–77,90]. Thus, changes in the TPCs of metabolism will induce changes in velocity TPCs, in turn affecting the ability of predators to find prey as well as the ability of those prey to escape. This relationship allows us to investigate how metabolic plasticity drives changes in species velocity and how the arising mismatches in trait performance may result in altered predator search rates using published mechanistic models[13,40,91].

**Metabolic rate.** The temperature dependence of metabolism was modelled as the modified Sharpe-Schoolfield equation[92]:

$$B = \frac{B_0 m^\beta e^{\frac{-E}{k}\left(\frac{1}{T} - \frac{1}{T_{ref}}\right)}}{1 + e^{\frac{E_d}{k}\left(\frac{1}{T_{pk}} - \frac{1}{T}\right)}}. \tag{1}$$

Here, $B$ is oxygen consumption rate, $B_0$ is the normalisation constant (rate at the reference temperature $T_{ref}$), $m$ is body mass, $\beta$ is the mass-dependent scaling factor of oxygen consumption, $E$ is the activation energy, $E_d$ is its deactivation energy, $k$ is Boltzmann's constant, $T$ is acute temperature (the experimental temperature in the vial for each $B$ measurement) in Kelvin and $T_{pk}$ is the temperature at which $B$ is maximised. We expect baseline performance ($B_0$), normalised to the mean temperature of a species' local habitat ($T_{ref}$) to reflect the population's biologically relevant thermal adaptation to that temperature.

**Statistics and reproducibility**. The Sharpe-Schoolfield equation (Eq. 1) was fitted to the respiration data across acute temperatures for each site-specific population with non-linear least squares (NLLS), using the nls.multstart package in R[93]. The starting parameters for each NLLS algorithm run were sampled 10,000 times from a normal distribution centred on values estimated from an Arrhenius plot[93]. The best-fit model out of 10,000 runs in each case was retained to calculate the estimates for $E$, $E_d$, $B_0$, $\beta$ and $T_{pk}$. The estimated parameter values from each TPC were then compared between populations. Each model fit was bootstrapped 10,000 times to obtain uncertainty bounds around the TPCs. From the bootstrapped predictions, the 97.5$^{th}$ and 2.5$^{th}$ quantiles were also calculated to produce 95% confidence interval bounds.

**Velocity**. The energetics of animal movement, and specifically swimming, have been extensively studied for all major vertebrate classes[75–77], but very few studies have focused on the energetics of invertebrate swimming[94–96]. Indeed, literature on invertebrate swimming is limited and we were unable to find any published works specifically looking at energetics of swimming in freshwater macroinvertebrates as studied here. Nevertheless, both vertebrate and invertebrate locomotion have been linked to metabolic rate increases and several empirical studies have measured the change in oxygen consumption with body velocity[55,97–99]. Moreover, the literature on animal movement suggests that velocity scales positively and linearly with metabolic rate[77,100]. Therefore, there is a relationship between cost of transport ($C$), the amount of energy (in joules (J)) needed to transport 1 Newton (N) over 1 m, for submerged swimmers, and metabolic rate ($B$). Rearranging Videler's[75,76] equation for velocity yields:

$$v = \frac{B\gamma}{Cmg} \quad (2)$$

Here, $B$ is oxygen consumption rate expressed in mmol.L$^{-1}$.mg$^{-1}$ and converted into J.s$^{-1}$ via $\gamma$ the conversion coefficient of oxygen combustion into energetic output, $m$ is mass in kg, $g$ is gravitational acceleration in m.s$^{-2}$, $v$ is velocity in m.s$^{-1}$, $C$ is expressed in J.N.m$^{-1}$ and was calculated from: $C = 1.1$ m$^{-0.38}$ (Videler, 1993). We estimated the energy allocated to muscle contraction and movement from the combustion of nutrients under aerobic (oxygen replete) conditions (Supplementary Methods 2[101,102]). The temperature dependence of $v$ is captured in the $B$ term (from Eq. 1).

**Search rate**. Having calculated velocities from oxygen consumption rate, we then investigated the strength of predator-prey interactions by calculating search rate: the rate at which a predator 'clears' (detects) an area or volume while searching for prey. Because search rate limits predator-prey encounter (and therefore consumption) rate when resources are relatively rare (arguably the most common scenario in field conditions), it strongly governs consumer-resource dynamics[40,91]. Search rates are related to the dimensionality of the environment (i.e., 2D or 3D) as:

$$a = v_r D \quad (3)$$

where $v_r$ is the relative velocity of a predator and its prey and $D$ is the dimensionality of the detection region during the search. Velocity is temperature-dependent[13], and the temperature-dependence of relative velocity for two individuals of different species moving randomly in an environment (i.e., active capture: predator foraging on an active moving prey) can be incorporated into Eq. 4 using Eq. 3, for both the predator and prey species (Supplementary Methods 2), giving:

$$a = \sqrt{\left(v_{0_r} m_r^{\beta_r} e^{\frac{-E_r}{k}\left(\frac{1}{T}-\frac{1}{T_{ref}}\right)}\right)^2 + \left(v_{0_c} m_c^{\beta_c} e^{\frac{-E_c}{k}\left(\frac{1}{T}-\frac{1}{T_{ref}}\right)}\right)^2} \, D. \quad (4)$$

Here, $v_0$ is the normalisation constant for velocity (equation S4) at a reference temperature ($T_{ref}$), set here at the mean temperature for each site,

$m$ is mass, $\beta$ is the mass scaling exponent, $E$ is activation energy, $T$ is temperature (in Kelvin (K)), $k$ is Boltzmann's constant and $c$ or $r$ subscripts correspond to predator (consumer) and prey (resource) parameters respectively. For benthic species such as *Chironomus* spp. a more appropriate equation is obtained by setting the prey's velocity to 0 (i.e., sessile prey: predator foraging on a static prey):

$$a = v_{0,c} m_c^{\beta_c} e^{\frac{-E_c}{k}\left(\frac{1}{T}-\frac{1}{T_{ref}}\right)} D \quad (5)$$

To account for the dimensionality of the environment (benthic ~2D or pelagic ~3D) in which each prey species interacts with the predator, we modelled search rates separately in 2 or 3D environments as[40]:

$$2D: a = v_r 2d_0 (m_c m_r)^{p_d} \quad (6)$$

$$3D: a = v_r \pi d_0 \left((m_c m_r)^{p_d}\right)^2 \quad (7)$$

where $d_0$ is the minimum reaction distance, $m_c$ and $m_r$ are average consumer and resource body mass respectively, and $p_d$ is the scaling exponent of mass with dimensionality. The values of $p_d$ in 2D and 3D were empirically derived from the only meta-analysis available on dimensionality scaling and are equal to ~0.21 and ~0.2 respectively[40].

We examined the differences between modelled search rates for predators living in different thermal environments. We used an Arrhenius plot to estimate the biological parameters that describe search rates ($E$ and $B_0$) at each site and for each foraging strategy. We then compared these parameters statistically to identify adaptive or acclimatory changes in search rates in response to warmer environments.

**Reporting summary**
Further information on research design is available in the Nature Portfolio Reporting Summary linked to this article.

## Data availability
Data used to produce this manuscript can be accessed freely at: https://github.com/FlavAff/Locomotion-metabolism-and-acclimation (https://doi.org/10.5281/zenodo.11155130[103]).

## Code availability
All code can be accessed here: https://github.com/FlavAff/Locomotion-metabolism-and-acclimation (https://doi.org/10.5281/zenodo.11155130[103]).

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

## Acknowledgements

We acknowledge the funding support of the Royal Society (RG150320) to R.L.K. and S.P., and Imperial College London to F.A. and R.L.K. M.G.M. acknowledges support to Iberian Ponds Network from the Portuguese Science and Technology Foundation (TrophicResponse: PTDC/BIA-BIC/0352/2014) and from the European Union's Horizon 2020 research and innovation programme under grant Nos 731065 (AQUACOSM). Special thanks go to Katarzyna Sroczynska and Cátia Lucio Pereira for their support in collecting the data used in this study.

## Author contributions

F. Affinito, M.G. Matias, S. Pawar and R.L. Kordas conceived the ideas and designed methodology; F. Affinito, M.G. Matias and R.L. Kordas collected the data; F. Affinito analysed the data and developed the mathematical model; F. Affinito wrote the manuscript. M.G. Matias, S. Pawar and R.L. Kordas edited the manuscript. All authors contributed critically to the manuscript drafts and gave final approval for publication.

## Competing interests

The authors declare no competing interests.
