## [Peer Review File · Communications Biology]

Reviewers' comments:

Reviewer #1 (Remarks to the Author):

This paper studies the relationship between individual metabolic rate and predator-prey dynamics by using empirical data as well as theoretical models. The authors empirically measured respiration rates (proxy for metabolic rate) in terms of three insect taxa, and then they used theoretical model to predict the effect of temperature on predator-prey interaction. The authors suggest the important role of temperature. Variations in temperature results in the variation of metabolic rate, and the metabolic rate changes individual movement. Individual movement will decide whether a prey can avoid a predator, which can influence the predator-prey interaction. I found the research topic is interesting. However, I have two major concerns. First, the rationality of some default assumptions is not provided. For example, the authors believe that temperature can have an effect on the energy that can be allocated to individual movement. However, I am not convinced by the underlying relationship between respiration rate with movement. Second, the authors pay more attention to what they found, but the importance is not sufficiently discussed in an ecological context.

Minor comments

Line 36 Can you give more specific information about the “different, sometimes opposite, ways”?

Line 38-39 I don't think it is suitable for using the word “beginning”. Almost 30 years ago, metabolic theory of ecology has been proposed (Please see the WBE model, West et al., 1997). In such a case, how could you say research on this area is just begin?

Line 45 “though” or “through”?

Lines 44-50 The sentences are difficult to understand. First, it is unclear the relationship between environmental temperature and body temperature. It seems the authors, in default, believes that changing in environmental temperature results in the changing in body temperature and thus the variation of metabolic rate. However, how about homothermal animal? Second, a clear logic here is that high temperature results in high respiration rate, and high resource demands as well as the relationship with growth rates. However, its effect on fitness is not clear.

Lines 58-61 The meanings here are unclear.

Lines 66-78 This is the very important paragraph linking the next paragraph which introduce your scientific problems. However, the impression you give me is that one thing “modify” another, and then the second “modify” the third. What are the underlying ecological implications?

Lines 75-78 What is the difference between your data and previous empirical data. A related question is what is the contribution and novelty of your study?

Line 84 A definition of the term “thermal acclimation or adaptation” is lacking at the first time when it

appears in the paper.

Line 89 Effect of temperature on what?

Line 176 Should be explained at the context here.

Line 197 what the specific relationship this scaling exponent refers to?

Lines 216-219 The logic between this sentence and the following ones should be clear if you do not put this as research gap.

Line 223 It is not clear whether the linear relationship is positive or negative.

Line 292 I don't think the word "good " is good here.

Line 328 "were" should be "was"

Line 360 I think here you mean that *Chironomus* spp. moves to a habitat with low temperature. Then, what is the logic between this sentence and the next?

Reviewer #2 (Remarks to the Author):

The manuscript by Affinito et al utilises metabolic rate measurements conducted on three freshwater invertebrate taxa (one predator and two potential prey) collected from pond experiments spanning three sites differing in their mean annual temperature (14-17 °C) in Portugal and Spain. The metabolic data is then used to model the relative velocity and search rate (a proxy for interaction strength) of the two predator-prey pairs, one of which involves a largely sessile prey moving only in the benthos (2-dimensional) and the other a highly mobile prey that moves through the water column (3-dimensional). The authors find differences between the three sites in the average metabolic rate and thermal sensitivity of metabolism, which leads to a mismatch in the relative velocity of the 2D interaction (predators are always faster than their prey and thus exhibit stronger interactions), but not of the 3D interaction.

The concept of the study is excellent, and I like how it builds on recent work by two of the authors on metabolic plasticity (in *Nature Communications*) to show the trophic consequences of that organismal ability to modulate metabolism in warmer environments. There is lots to like in here that could fuel further studies and empirical validation of the modelling conducted. However, my biggest concern at

present is the completely theoretical nature of the trophic component, which builds from a single (non-trophic) trait without any empirical sense-check if the predictions on relative velocities and search rates are accurate.

I was fully expecting to see some empirical measurement of velocity and/or search rate using video-tracking equipment or functional response experiments, which of course are time-consuming but standard approaches in literature on this subject. That would have given a far more empirically grounded comparison of the trade-off between temperature effects on energy intake and energy expenditure. Instead, the conclusions drawn about temperature effects on mismatches between relative velocities and interaction strengths between predators and prey are entirely inferred from metabolism, which may not prove to be the truth given the likely emergent effects that occur across different biological traits, each underpinned by varying contributions of physiology (e.g. metabolism is not really under the organism's control) and cognition (e.g. movement and feeding are subject to behavioural decisions).

I see no obvious way to fix this unless the authors happened to conduct feeding rate experiments in parallel, e.g. there is some mention of feeding trials, but perhaps this was not extensive enough to allow an empirical validation of the modelling? There may still be enough here to stimulate further research on the topic, however, and I am broadly supportive of the approach. I have some additional minor comments below to help improve the study.

Specific comments:

Ln45. Increasing temperature doesn't always lead to an elevated respiration rate, e.g. if the organism is experiencing temperatures beyond its thermal optimum. Change the language here to reflect that.

Ln108. I guess a key assertion of the manuscript is that the temperature differences between sites underpin most of the differences in the thermal responses of the constituent organisms. That requires some confidence that there were not other parameters apart from temperature that differed between sites though, e.g. a supplementary table showing that nutrient concentrations and other important chemical variables were approximately overlapping across sites. Even the temperature differences are not that great (particularly between Porto and Evora), however, which suggests that site-level contingencies may be more at play here.

Ln165. It sounds odd to say that oxygen was "not allowed to" drop below 70%, as if you could somehow tell the organism to stop breathing if it was getting close to that. I assume you mean that oxygen "never dropped below" 70% during all experiments (in which case it might help to give the minimum oxygen concentration across all experiments), or that you excluded any experiments where it dropped below 70% (and give the number of times that happened).

Ln300. There is no Figure 1e.

Ln309. The results in this paragraph are interesting, but ultimately a modelling exercise that could really have benefited from some empirical validation, e.g. video tracking to quantify the relative velocities. The assumption of the model is that these differences in relative velocities are entirely underpinned by

metabolism, but there could be individual decisions that could over-ride those, e.g. choosing to move faster/slower given the energetic need or predation pressure to do so, even if that could be physiologically restricted to a short burst followed by a later period of inactivity to recover.

Ln386. I like this explanation of why the mobile predator-prey pair may track each other's TPCs (the prey typically lack refugia in the water column) whereas the sessile prey can bury into the sediment, meaning it can overcome its slower pace with warming relative to the predator. Ultimately though, there is only one predator-prey example of each type of interaction, making it difficult to know if this sort of generalisation is really the case.

Ln408. Can we really know if the predictions are accurate without some empirical ground-truthing of the approach though?

Figure 2. The modelled curves look like a poor fit to the data points in each of these figures. For example, the red confidence intervals in panel b stretch upwards well beyond any of the red data points and yet omit a large proportion of the red data points below. The same is true of the blue confidence intervals and data points in panels a and c. Are these really the correct model fittings? If so, then the models seem to do a poor job of predicting the data (which seems odd given the high r^2 values reported at Ln292). It is also generally difficult to distinguish the colour of the intermediate temperature from the other two. I suggest using something much clearer, e.g. yellow, and also different plotting symbols, e.g. circles, triangles, and squares. A bit of jitter along the x-axis might also help to identify the various data points.

Figure 5. The purpose of the rectangular blocks of colour is unclear and not explained in the figure legend or main text. It seems odd to conflate species identity with prey strategy, i.e. the curves should EITHER be distinguished based on prey identity OR by the 2D/3D characteristic, but not both. My preference would be for prey identity since you would need multiple predator-prey pairs to generalise to overall 2D and 3D strategies. Why are there no confidence intervals around the predicted search rates? It is unclear if there would really be any difference between the two prey species without those. It is also unclear whether there is a significant difference between sites, e.g. consider one more panel, similar to Figure 3, which shows the B_0 and/or activation energy of search rate (with confidence intervals) for each site.

Affinito *et. al.* response to reviewers

We thank the editor for the opportunity to submit a revised manuscript. In addressing the reviewers' comments, we believe we have substantially improved the clarity of our findings.

We addressed reviewer one's main comments by providing more rationale for our approach and analysis and by giving more detail about the ecological relevance of our work at several points throughout the manuscript.

We addressed reviewer two's main comments by expanding upon the contribution to knowledge that our paper provides and the context within which we believe this piece fits. We more explicitly outline the limitations of our study and its strengths and respond to their concerns in detail. Specifically, we address the value of the modelling approach used and expand on its value within the field of theoretical ecology.

We have revised the manuscript to address the concerns of the reviewers and the essential revisions outlined by the editor. We have responded to each of the reviewers' concerns in our point-by-point response below.

Recommendations made by the reviewers are given in black, our responses are given in blue.

Reviewer #1

This paper studies the relationship between individual metabolic rate and predator-prey dynamics by using empirical data as well as theoretical models. The authors empirically measured respiration rates (proxy for metabolic rate) in terms of three insect taxa, and then they used theoretical model to predict the effect of temperature on predator-prey interaction. The authors suggest the important role of temperature. Variations in temperature results in the variation of metabolic rate, and the metabolic rate changes individual movement. Individual movement will decide whether a prey can avoid a predator, which can influence the predator-prey interaction.

I found the research topic is interesting. However, I have two major concerns. First, the rationality of some default assumptions is not provided. For example, the authors believe that temperature can have an effect on the energy that can be allocated to individual movement. However, I am not convinced by the underlying relationship between respiration rate with movement.

We thank the reviewer for their insightful comments. Throughout the manuscript, we have elaborated on the rationale for how temperature influences movement, emphasizing its pivotal role in biological processes. Specifically, we direct your attention to lines 42 to 49 where we highlight the established relationship between temperature and metabolism and to lines 193 to 201 where we have now expanded on the link between metabolism and movement. To summarize, the core of our argument hinges on how temperature modulates metabolic rates, thereby influencing ATP availability essential for all forms of cellular activity, including movement. ATP is required for all cellular processes, including movement. literature on movement in fact highlights the relationship between energy and various locomotion types, including swimming. Therefore, temperature by affecting respiration rates will also affect individual movement. We model this using mechanistic models of respiration and movement available in the literature. Nevertheless, we acknowledge that significant uncertainty exists in the link between respiration and body velocity. We have addressed

this in the main manuscript and added several key caveats in response to specific comments from both reviewers below.

Second, the authors pay more attention to what they found, but the importance is not sufficiently discussed in an ecological context.

We would like to point out lines 377 to 385 and lines 402-406 where we specifically discuss the ecological implications of our findings for the predator-prey relationships studied here. Additionally, we have expanded upon the ecological implications of our results in freshwater systems in the summary section of the manuscript lines 468-476.

Minor Comments

Line 36 Can you give more specific information about the “different, sometimes opposite, ways”? We decided to remove the “sometimes opposite” statement here as we feel it is not the right place in the text to expand on differences in thermal adaptation/acclimation of species.

Line 38-39 I don't think it is suitable for using the word “beginning”. Almost 30 years ago, metabolic theory of ecology has been proposed (Please see the WBE model, West et al., 1997). In such a case, how could you say research on this area is just begin?

The reviewer here is correct, we would like to explain that we were not meaning that MTE is only just beginning, rather that temperature-related mechanistic understanding of traits is still emerging in the literature and there remains much to understand from this perspective. To avoid any confusion, we have removed the word “beginning” and edited the sentence to say that this mechanistic understanding “remains to be developed”.

Line 45 “though” or “through”?

Correct, edited.

Lines 44-50 The sentences are difficult to understand. First, it is unclear the relationship between environmental temperature and body temperature. It seems the authors, in default, believe that changing in environmental temperature results in the changing in body temperature and thus the variation of metabolic rate. However, how about homothermal animal? Second, a clear logic here is that high temperature results in high respiration rate, and high resource demands as well as the relationship with growth rates. However, its effect on fitness is not clear.

Thank you for this comment, we have now specified that the change in environmental temperature results in a changing of body temperature *for ectotherms* (the subjects of our study). Additionally, we split the last sentence in two to specify that changes in resource demand affect predation & growth. Then changes in resource acquisition (e.g. through predation) will affect survivability and reproduction chances therefore affecting individual fitness outcomes. See lines 44-52.

Lines 58-61 The meanings here are unclear.

We have rephrased line 59 to clarify the sentence and explain that thermal sensitivity changes as the TPC becomes steeper and then explaining how this can happen biologically.

Lines 66-78 This is the very important paragraph linking the next paragraph which introduce your scientific problems. However, the impression you give me is that one thing “modify” another, and then the second “modify” the third. What are the underlying ecological implications?

We have added a few sentences expanding on the ecological implications of mismatches in thermal performance between prey and predators. Lines 72-80 now provide some ecological impacts of these metabolic mismatches driven by the effect of warming.

Lines 75-78 What is the difference between your data and previous empirical data. A related question is what is the contribution and novelty of your study?

Our data is not per se different from other measurements of oxygen consumption, the novelty of our study is in the combination of such empirical measurements with biomechanical and consumer-resource theory mechanistic models to predict the effects of changes in basal metabolic rate on species interactions. We have attempted to make this point clearer in the following paragraph where we believe it to be more appropriate as it presents our work explicitly. See lines 91-103.

Line 84 A definition of the term “thermal acclimation or adaptation” is lacking at the first time when it appears in the paper.

We now provide a short definition of the terms in brackets and refer the reader to a more in-depth explanation “phenotypic changes resulting from an organism coping with a new thermal regime or due to selection of better thermally adapted phenotypes in a population (Angilletta, 2009)”

Line 89 Effect of temperature on what?

Edited to include “on traits such as respiration, locomotion and predator search rate”

Line 176 Should be explained at the context here.

While we understand the reviewer's suggestion to add more context in this section (now line 190), we respectfully offer a different perspective. This first statement simply introduces the modelling section for which the context has been set out in the introduction. This context has also been expanded in the introduction (see previous comments) to explain the study better. The sentence in question simply states what the *Modelling* section sets out to describe (“mechanistic models of metabolism, movement and predation”) and why (to estimate “the temperature dependence of unmeasured velocities and search rates for our species *from measured respiration rates*” – section in italics was added to help clarify the context of this sentence). We believe that delving further into the context of the study here would add confusion to this section that focuses on the models used in the study.

Line 197 what the specific relationship this scaling exponent refers to?

“ β is the mass-dependent scaling factor of oxygen consumption” now specified in text.

Lines 216-219 The logic between this sentence and the following ones should be clear if you do not put this as research gap.

The logic between sentences has been improved here to highlight that there is indeed a research gap when it comes to the study of the energetics of movement in freshwater macroinvertebrates but that we are relying on the more extensive works published on the energetics of movement in other species and the oxygen consumption of movement. See lines 238-249.

Line 223 It is not clear whether the linear relationship is positive or negative.

We have specified the relationship as positive.

Line 292 I don't think the word “good ” is good here.

Agreed, edited.

Line 328 “were” should be “was”

Correct, edited.

Line 360 I think here you mean that *Chironomus* spp. moves to a habitat with low temperature.

Then, what is the logic between this sentence and the next?

Here we mean that *Chironomus* spp. cannot adapt to warming as well as its predator and therefore may become a more preferred prey item as it is easier to find. The next sentence makes explicit that under warming these prey’s inability to adapt will make them easier to catch as they are not able to move as well. To help clarify the logic between both sentences, we have changed “move” to “swim” as we are referring to the reduction in the prey’s ability to move within its environment (i.e. to swim).

Reviewer #2

The manuscript by Affinito et al utilises metabolic rate measurements conducted on three freshwater invertebrate taxa (one predator and two potential prey) collected from pond experiments spanning three sites differing in their mean annual temperature (14-17 °C) in Portugal and Spain. The metabolic data is then used to model the relative velocity and search rate (a proxy for interaction strength) of the two predator-prey pairs, one of which involves a largely sessile prey moving only in the benthos (2-dimensional) and the other a highly mobile prey that moves through the water column (3-dimensional). The authors find differences between the three sites in the average metabolic rate and thermal sensitivity of metabolism, which leads to a mismatch in the relative velocity of the 2D interaction (predators are always faster than their prey and thus exhibit stronger interactions), but not of the 3D interaction.

The concept of the study is excellent, and I like how it builds on recent work by two of the authors on metabolic plasticity (in *Nature Communications*) to show the trophic consequences of that organismal ability to modulate metabolism in warmer environments. There is lots to like in here that could fuel further studies and empirical validation of the modelling conducted. However, my biggest concern at present is the completely theoretical nature of the trophic component, which builds from a single (non-trophic) trait without any empirical sense-check if the predictions on relative velocities and search rates are accurate.

I was fully expecting to see some empirical measurement of velocity and/or search rate using video-tracking equipment or functional response experiments, which of course are time-consuming but standard approaches in literature on this subject. That would have given a far more empirically grounded comparison of the trade-off between temperature effects on energy intake and energy expenditure. Instead, the conclusions drawn about temperature effects on mismatches between relative velocities and interaction strengths between predators and prey are entirely inferred from metabolism, which may not prove to be the truth given the likely emergent effects that occur across different biological traits, each underpinned by varying contributions of physiology (e.g. metabolism is not really under the organism’s control) and cognition (e.g. movement and feeding are subject to behavioural decisions).

I see no obvious way to fix this unless the authors happened to conduct feeding rate experiments in parallel, e.g. there is some mention of feeding trials, but perhaps this was not extensive enough to

allow an empirical validation of the modelling? There may still be enough here to stimulate further research on the topic, however, and I am broadly supportive of the approach. I have some additional minor comments below to help improve the study.

We are grateful for the comments provided here and below and fully understand the reviewer's perspective. We would have liked to combine our work with empirical measurements of predator search rates but were unable to conduct sufficient experiments during the field season to add to this publication. Empirical work for quantifying encounter and consumption rates in nature is a difficult task (Dell *et al.*, 2014) which future work should focus on resolving. Therefore, we do indeed rely on predictions inferred from metabolism measurements. One of the main contributions of our paper is to further theoretical and empirical work on the role of metabolic plasticity. Indeed, the Kordas *et al.*, (2022) paper that the Reviewer cites also does not actually quantify trophic interactions directly but relies on metabolic theory to predict ecosystem level dynamics. As such, the issue of plasticity (García *et al.*, 2023: <https://doi.org/10.1038/s41564-022-01283-w>) is pervasive across organismal groups in ecosystems, and we hope that future work will build on our study to validate or refute these predictions in an effort to further our mechanistic understanding of the effects of temperature.

In summary, our approach provides a blueprint for work combining mechanistic metabolism models with movement ecology and consumer-resource theory and points to the importance of combining empirical measurements with ecological theory. As the field of movement ecology expands, its integration with the mechanistic effects of temperature—especially in the context of escalating climate change impacts—becomes increasingly vital, as our results suggest. Importantly, our theoretical prediction results align well with what can be expected in nature and we present this in our discussion where we refer to genetic experiments of chironomid thermal adaptation and the ecology of mayflies and dragonflies. As such, our approach yields believable results that are promising and should spur the community forward in exploring the direct links between temperature, metabolism, movement and species interactions. We trust that our responses and the revisions made to the manuscript adequately address the concerns raised.

Specific comments:

Ln45. Increasing temperature doesn't always lead to an elevated respiration rate, e.g. if the organism is experiencing temperatures beyond its thermal optimum. Change the language here to reflect that. Correct, we added "unless the temperature increases beyond an organism's thermal optimum" to the sentence.

Ln108. I guess a key assertion of the manuscript is that the temperature differences between sites underpin most of the differences in the thermal responses of the constituent organisms. That requires some confidence that there were not other parameters apart from temperature that differed between sites though, e.g. a supplementary table showing that nutrient concentrations and other important chemical variables were approximately overlapping across sites. Even the temperature differences are not that great (particularly between Porto and Evora), however, which suggests that site-level contingencies may be more at play here.

The reviewer makes an important point. We would like to emphasise that the mesocosms in the Iberian Peninsula have been set up to minimise environmental heterogeneity whilst allowing for natural thermal fluctuations. We now address this issue on lines 125-142 of the main text. In short, we explain that all mesocosms are made of the same material and were seeded in identical ways. As such, all mesocosms were oligotrophic from the start with no nutrient additions, maintaining the same pH and salinity as in the naturally occurring ponds nearby. The biggest difference between

regions is likely to be temperature and rainfall as the experiment was set up intentionally for these reasons. Rainfall may have affected nutrient concentrations, but the availability of nutrients is more likely to affect population abundance than the oxygen consumption rate of individuals. As we are focused on metabolic constraints on species interactions, and rates, and metabolic rates themselves are most strongly governed by size-scaling laws and temperature-driven enzyme kinetics, as shown by our study, even small variation in temperature between sites creates sufficient fitness differences to drive plastic metabolic (respiratory) responses of the organisms. We provided some additional context to this point in lines 124-129.

Ln165. It sounds odd to say that oxygen was “not allowed to” drop below 70%, as if you could somehow tell the organism to stop breathing if it was getting close to that. I assume you mean that oxygen “never dropped below” 70% during all experiments (in which case it might help to give the minimum oxygen concentration across all experiments), or that you excluded any experiments where it dropped below 70% (and give the number of times that happened).

Sentence has been modified to reflect that we used the 70% threshold as a way to end trials.

Concentrations can be found in the dataset and minimum oxygen concentration across experiments was 107 $\mu\text{mol/L}$

Ln300. There is no Figure 1e.

Correct, edited

Ln309. The results in this paragraph are interesting, but ultimately a modelling exercise that could really have benefited from some empirical validation, e.g. video tracking to quantify the relative velocities. The assumption of the model is that these differences in relative velocities are entirely underpinned by metabolism, but there could be individual decisions that could over-ride those, e.g. choosing to move faster/slower given the energetic need or predation pressure to do so, even if that could be physiologically restricted to a short burst followed by a later period of inactivity to recover. This is true, to avoid any confusion, we have edited the text to specify that these are indeed “predicted” results. Direct measurements of body velocities were outside the scope of what we were able to do in this study. Empirical data on the movement of freshwater macroinvertebrates are largely absent in the literature due to the difficulty of automated tracking in such environments (Dell *et al.*, 2014: <https://doi.org/10.1016/j.tree.2014.05.004>). We were conscious of the potential inaccuracies and biases in using biomechanical theory to link metabolism and locomotion, which is why we made sure that we did due diligence in terms of propagating uncertainty through the two levels of the modelling (Figures 4-5). However, the effect of temperature on biological rates (through its effect on metabolism) is undeniable and found throughout the literature on predation, growth rates, ecosystem function and movement. Thus, these results provide a set of testable predictions that will help shape future research effort on the effects of warming as well draw attention to the need to better quantify the key rates and traits underlying species interactions in aquatic ecosystems. We also now address this point more directly in lines 450-462 of the revised manuscript.

Ln386. I like this explanation of why the mobile predator-prey pair may track each other's TPCs (the prey typically lack refugia in the water column) whereas the sessile prey can bury into the sediment, meaning it can overcome its slower pace with warming relative to the predator. Ultimately though,

there is only one predator-prey example of each type of interaction, making it difficult to know if this sort of generalisation is really the case.

Ln408. Can we really know if the predictions are accurate without some empirical ground-truthing of the approach though?

True, this may indeed only be relevant to this specific case of interacting species. Please note that we are in fact not generalising in this section (lines 379-390) and only talking about the *S. striolatum* – *Chironomus spp.* interacting pair. Additionally, this section speaks to reviewer 1's comments on the ecological context of our work. As such our study takes a key step towards developing theory and identifying the data needed to allow a more general... as we have highlighte in the previous response (also see lines xx-xx of the revised manuscript).

As we explain in the comment for Ln309, we agree with the reviewer that these predictions can indeed not be verified without empirical ground truthing. Nevertheless, we believe they are a valuable contribution to the literature on the thermal plasticity of species and on the mechanistic processes underpinning it. We include some of the points raised in our comment in lines 427-440.

Figure 2. The modelled curves look like a poor fit to the data points in each of these figures. For example, the red confidence intervals in panel b stretch upwards well beyond any of the red data points and yet omit a large proportion of the red data points below. The same is true of the blue confidence intervals and data points in panels a and c. Are these really the correct model fittings? If so, then the models seem to do a poor job of predicting the data (which seems odd given the high r^2 values reported at Ln292). It is also generally difficult to distinguish the colour of the intermediate temperature from the other two. I suggest using something much clearer, e.g. yellow, and also different plotting symbols, e.g. circles, triangles, and squares. A bit of jitter along the x-axis might also help to identify the various data points.

Thank you for your edit suggestions, we have made changes that we hope are helpful (colour to yellow, shapes and jitter). Regarding model fit, please note that this amount of spread and uncertainty is typical of thermal performance curves for metabolic rates, even with high R^2 values. Here are some example papers with similar looking curves:

Sohlström et al 2021 <https://doi.org/10.1111/gcb.15715>

Kordas et al 2022 <https://www.nature.com/articles/s41467-022-29808-1>

Rall et al 2010 <https://doi.org/10.1111/j.1365-2486.2009.02124.x>

Clarke 2006 <https://www.jstor.org/stable/3806578>

Cloyed et al 2019 <https://doi.org/10.1111/1365-2656.12976>

González-Ferreras et al 2023 <https://www.nature.com/articles/s41467-023-43478-7>

Figure 5. The purpose of the rectangular blocks of colour is unclear and not explained in the figure legend or main text. It seems odd to conflate species identity with prey strategy, i.e. the curves should EITHER be distinguished based on prey identity OR by the 2D/3D characteristic, but not both. My preference would be for prey identity since you would need multiple predator-prey pairs to generalise to overall 2D and 3D strategies. Why are there no confidence intervals around the predicted search rates? It is unclear if there would really be any difference between the two prey species without those. It is also unclear whether there is a significant difference between sites, e.g. consider one more panel, similar to Figure 3, which shows the B_0 and/or activation energy of search rate (with confidence intervals) for each site.

The rectangular boxes correspond to the operational temperature range (originally mentioned only in the legend of Figure 4 now also in Figure 5).

We have removed the prey strategy from the legend as we recognise it causes confusion. They were included only to mention which model was used for each prey but are not relevant to the interpretation of the graph, especially as we are not comparing between strategies as mentioned in the comment. Therefore, curves were already distinguished by prey identity.

We have added confidence intervals.

We opted against adding a panel as figure 3 because the activation energies (representing thermal sensitivity) discussed in text have very small confidence intervals and significantly different values (see Table S3) which are already captured by the steepness of the curves in figure 5. Therefore, adding another panel creates more information to process but in our opinion adds little value.

REVIEWERS' COMMENTS:

Reviewer #1 (Remarks to the Author):

The authors have addressed my concerns, and I have no more comments.

Reviewer #2 (Remarks to the Author):

The revised manuscript and response document do a satisfactory job of addressing the concerns of the two reviewers. I appreciate the new sentences in response to reviewer 1's comments, providing greater justification for the link between metabolism and movement and the ecological context of their findings. Note that the line numbers mentioned in the response document do not correspond to altered text in the manuscript and there were no track changes, so it was challenging to follow these edits. I suggest including direct quotes to revised text in any future response documents to avoid such issues.

I remain sceptical about my major concern surrounding the lack of empirical validation of the model predictions, which has not been formally addressed in the manuscript, but I accept the authors' argumentation about why this was complicated and the value that their study still provides. I suggest adding a paragraph of discussion acknowledging this issue and identifying the need for empirical validation of their predictions using approaches like video-tracking and functional response experiments. This will at least more explicitly point the way for follow-up work to validate their conclusions.

Affinito *et. al.* response to reviewers

We thank the editor for the opportunity to submit a final edit of our manuscript for publication in Communications Biology.

There were no comments to address for reviewer 1.

We addressed reviewer two's remaining comment by including a paragraph discussing the importance of future works focusing on empirical validation of our work using relevant experimental tools such as functional response and video tracking experiments.

We have revised the manuscript to address the remaining concern of the reviewer and the essential revision outlined by the editor. We have added a section discussing empirical validation in the main manuscript and included here in blue.

Reviewer #1

The authors have addressed my concerns, and I have no more comments.

Reviewer #2

The revised manuscript and response document do a satisfactory job of addressing the concerns of the two reviewers. I appreciate the new sentences in response to reviewer 1's comments, providing greater justification for the link between metabolism and movement and the ecological context of their findings. Note that the line numbers mentioned in the response document do not correspond to altered text in the manuscript and there were no track changes, so it was challenging to follow these edits. I suggest including direct quotes to revised text in any future response documents to avoid such issues.

We apologise for the line number error and have made sure to include a file with tracked changes this time and a direct quote of our edit below.

I remain skeptical about my major concern surrounding the lack of empirical validation of the model predictions, which has not been formally addressed in the manuscript, but I accept the authors' argumentation about why this was complicated and the value that their study still provides. I suggest adding a paragraph of discussion acknowledging this issue and identifying the need for empirical validation of their predictions using approaches like video-tracking and functional response experiments. This will at least more explicitly point the way for follow-up work to validate their conclusions.

We understand the reviewer's reservations about the lack of empirical validation and are grateful for the suggestion to address this further in the discussion. We have now added a paragraph after the commentary on future works. We copy this paragraph below:

"Time constraints prevented us from empirically testing our predictions about the consequences of warming and plasticity on predator-prey performance mismatches and the emergent temperature-dependence of interaction rates. Testing predictions of such mechanistic theory for temperature-dependent predator-prey interactions would require functional response experiments using the same species-pair and site combinations¹⁻⁴. The search rates estimated from these experiments could then be compared with the predicted values (Figure 5). However, detecting acclimatory changes would

require a higher level of replication to estimate the thermal plasticity of search rate parameters. Furthermore, the predictions of the model for the temperature-dependence of velocity would also need to be tested (Figure 4). Direct measurement of body velocity can be done using video tracking powered by machine learning⁵⁻⁹. This is challenging because of the difficulty of automated tracking in such environments, which likely explains the paucity of existing data on the movement of freshwater macroinvertebrates^{10,11}. Lab experiments with wild-caught organisms may help address this gap^{12,13} provided the challenge of recreating the natural thermal and structural environment can be overcome.”

1. Rall, B. C., Vucic-Pestic, O., Ehnes, R. B., Emmerson, M. & Brose, U. Temperature, predator–prey interaction strength and population stability. *Global Change Biology* **16**, 2145–2157 (2010).
2. Englund, G., Öhlund, G., Hein, C. L. & Diehl, S. Temperature dependence of the functional response. *Ecology Letters* **14**, 914–921 (2011).
3. Vucic-Pestic, O., Ehnes, R. B., Rall, B. C. & Brose, U. Warming up the system: higher predator feeding rates but lower energetic efficiencies. *Global Change Biology* **17**, 1301–1310 (2011).
4. Archer, L. C. *et al.* Consistent temperature dependence of functional response parameters and their use in predicting population abundance. *Journal of Animal Ecology* **88**, 1670–1683 (2019).
5. Patullo, B. W., Jolley-Rogers, G. & Macmillan, D. L. Video tracking in the extreme: Video analysis for nocturnal underwater animal movement. *Behavior Research Methods* **39**, 783–788 (2007).
6. Conklin, E. E., Lee, K. L., Schlabach, S. A. & Woods, I. G. with Open Source Software and Off-the-Shelf Video Equipment. *JUNE* **13**, 120–125 (2015).
7. Sridhar, V. H., Roche, D. G. & Giggins, S. Tracktor: Image-based automated tracking of animal movement and behaviour. *Methods in Ecology and Evolution* **10**, 815–820 (2019).
8. Crall, J. D., Gravish, N., Mountcastle, A. M. & Combes, S. A. BEEtag: A Low-Cost, Image-Based Tracking System for the Study of Animal Behavior and Locomotion. *PLoS ONE* **10**, e0136487 (2015).

9. Panadeiro, V., Rodriguez, A., Henry, J., Wlodkovic, D. & Andersson, M. A review of 28 free animal-tracking software applications: current features and limitations. *Lab Anim* **50**, 246–254 (2021).
10. Francisco, F. A., Nührenberg, P. & Jordan, A. High-resolution, non-invasive animal tracking and reconstruction of local environment in aquatic ecosystems. *Mov Ecol* **8**, 27 (2020).
11. Dell, A. I. *et al.* Automated image-based tracking and its application in ecology. *Trends in Ecology & Evolution* **29**, 417–428 (2014).
12. Pérez-Escudero, A., Vicente-Page, J., Hinz, R. C., Arganda, S. & De Polavieja, G. G. idTracker: tracking individuals in a group by automatic identification of unmarked animals. *Nat Methods* **11**, 743–748 (2014).
13. Ioannou, C. C., Guttal, V. & Couzin, I. D. Predatory Fish Select for Coordinated Collective Motion in Virtual Prey. *Science* **337**, 1212–1215 (2012).